# The Effect of Oral Diet Training in Indwelling Nasogastric Tube Patients with Prolonged Dysphagia

**DOI:** 10.3390/nu16152424

**Published:** 2024-07-26

**Authors:** Byung-chan Choi, Sook Joung Lee, Eunseok Choi, Sangjee Lee, Jungsoo Lee

**Affiliations:** Department of Physical Medicine and Rehabilitation, College of Medicine, The Catholic University of Korea, Seoul 06591, Republic of Korea; hjsrh@naver.com (B.-c.C.); choies612@gmail.com (E.C.); upperlimb@hanmail.net (S.L.); drlee71@naver.com (J.L.)

**Keywords:** nasogastric tube, deglutition disorder, chronic dysphagia, oral training

## Abstract

**Background:** Patients with severe dysphagia are usually fed using a nasogastric tube (NGT). Many patients who receive long-term NGT feeding are unable to obtain sufficient nutrients orally immediately after NGT removal. Thus, a transitional period involving oral diet training is required to transition from NGT feeding to exclusive oral feeding. We aimed to investigate the therapeutic effect of oral diet training in indwelling NGT patients with prolonged dysphagia. **Methods:** A total of 175 patients who were fed using an NGT for more than 4 weeks were enrolled. Their swallowing function was evaluated by a videofluoroscopic swallowing study (VFSS). During the VFSS, patients received thick and thin barium while the NGT was inserted. Then, the patients underwent a VFSS without an NGT thirty minutes after NGT removal. If a patient had no aspiration with NGT inserted during the VFSS, oral diet training combined with NGT feeding was recommended. **Results:** Of the 49 indwelling NGT patients who were recommended to receive oral diet training, 39 (79.6%) transitioned to exclusive oral feeding. A transition period of 2–8 weeks was required for them to achieve full oral feeding. Patients who were eligible for oral feeding trials showed no significant aspiration during the VFSS with an NGT inserted and had sufficient cough function. Patients who required prolonged NGT feeding and who could not complete oral trials showed significant aspiration during the VFSS when an NGT was inserted. **Conclusions:** This study demonstrated that oral diet training combined with NGT feeding is safe in patients with prolonged dysphagia who have sufficient cough function and no aspiration during VFSS. We suggest that if the patient is a proper candidate for NGT removal, direct oral feeding training with an NGT inserted could be a useful therapeutic strategy during the transitional period from long-term NGT feeding to successful oral feeding.

## 1. Introduction

Patients with severe swallowing difficulties usually use compensatory feeding methods such as nasogastric tube (NGT) feeding or percutaneous endoscopic gastrectomy (PEG) [1,2,3]. NGT feeding may be an appropriate alternative type of enteral feeding method for patients who are unable to obtain adequate nutrition via an oral route. However, some patients are fed using an NGT for relatively long periods, which may have a negative impact on their swallowing function [4]. Long-term NGT indwelling may cause patients to gradually lose the ability to chew and swallow. As a result, the stimulation the brain receives decreases, and the incidence of cognitive impairment increases [5]. Furthermore, prolonged dysphagia is strongly associated with poor functional outcomes after brain injury [6,7].

Long-term NGT placement can also cause complications such as gastroesophageal reflux or aspiration pneumonia, nasal wing lesions, and chronic sinusitis [8]. Although many studies and guidelines recommend PEG tube insertion in patients with prolonged dysphagia for more than 4–6 weeks [8,9,10], the appropriate patients and timing for this intervention are controversial. Furthermore, there are many patients with prolonged NGT insertion who do not want to undergo PEG tube insertion for various reasons [11,12].

Notably, passing a swallowing test does not guarantee the end of NGT feeding for dysphagic patients. Patients with prolonged NGT insertion can lose their oropharyngeal swallowing function [4], and many patients who receive NGT feeding for a long time are unable to obtain sufficient amounts of nutrients orally immediately after NGT removal [13]. There are also patients who require NGT reinsertion due to aspiration pneumonia or malnutrition. As patients with dysphagia have various etiologies and undergo different recovery processes, multiple and complex factors are associated with successful exclusive oral feeding. Frequent evaluation and appropriate management of dysphagia symptoms are very important strategies.

If the patient is a proper potential candidate for NGT removal, individualized and comprehensive approaches are needed to achieve full oral feeding. At this point, we would like to emphasize that a transitional period involving oral diet training is required to transition from long-term indwelling NGT feeding to exclusive oral feeding. However, the impact of long-term indwelling NGT insertion on oropharyngeal function remains largely unknown. Previous studies have shown various results regarding the effect of long-term NGT insertion on swallowing function [3,4,8].

This study aims to evaluate the therapeutic effect of oral diet training in indwelling NGT patients with prolonged dysphagia. We also attempt to identify factors that may be associated with safe oral feeding attempts with indwelling NGT.

## 2. Materials and Methods

This study was designed as a retrospective study. Dysphagic patients admitted to our institution between 1 February 2020 and 30 December 2022 were screened for this study. We reviewed patients’ medical charts, which were evaluated through a video-fluoroscopic swallowing study (VFSS) during the NGT feeding period. The study protocol was approved by the institutional review board of the Catholic University of Korea, Daejeon St. Mary’s Hospital (IRB No: DC22RISI0013). Because of the retrospective study design and minimal harm to the patients, informed consent was waived. All patient-specific identifiers were deleted from the data set before analysis.

### 2.1. Evaluations

Medical records of 175 patients with severe dysphagia who were fed via an NGT for more than 4 weeks were retrospectively reviewed. The swallowing function of all patients was evaluated by a VFSS. During the VFSS, patients received thick and thin barium while the NGT was insulted. Then, the patients’ NGTs were removed. Thirty minutes after NGT removal, the patients underwent a VFSS without an NGT. All VFSS processes were recorded, and three expert physicians who specialized in physical medicine and rehabilitation physiatrists interpreted the results. The diet type was determined based on the VFSS results.

The swallowing function of patients was measured using the functional dysphagia scale (FDS) and penetration aspiration scale (PAS) according to the VFSS result. The VFSS was performed with a modified Logerman protocol by physicians [14]. The FDS is a calculation system used to estimate the severity of dysphagia. This FDS is composed of 11 items, including 4 oral function items and 7 pharyngeal function items, which are observed during a VFSS [15]. A higher FDS indicates more severe dysphagia.

The PAS can evaluate airway invasion. The score was determined according to the depth to which food materials passed the vocal cord into the airway and whether the food materials entering the airway could be expectorated [16]. The laryngotracheal aspiration categories correspond to levels 6–8 on the PAS. A PAS score of 8 indicates that food materials enter the airway and pass below the vocal folds and that no effort is made to expectorate the material.

Cough function was measured using peak cough flowmetry (PCF) after the VFSS. Voluntary coughing power was measured by asking patients to cough as forcefully as possible using a peak flow meter. They were allowed to exert their maximal effort at least 3 times. The maximum value from the 3 trials was used for the statistical analysis. The PCF values are primary parameters widely used to estimate voluntary cough ability [17,18].

### 2.2. Patient Grouping

Patients were divided into one of the following 3 groups:

Group 1 included patients who could only consume food orally.

Group 2 included patients who could attempt oral feeding combined with NGT insertion. In this group, NGT feeding was the main feeding method. Patients in this group could intake soft and thick food orally but were not able to intake a sufficient amount of food orally; thus, oral feeding with an indwelling NGT was attempted.

Group 3 included patients who were not able to attempt oral feeding with an indwelling NGT.

Figure 1 shows the findings of the VFSS with an NGT inserted. Patients assigned to Group 2 had no aspiration when oral swallowing was attempted with an NGT inserted during the VFSS (Figure 1A). These patients were allowed an oral feeding trial with an NGT inserted. However, the patients assigned to Group 3 showed aspiration when they tried to swallow orally with an NGT inserted (Figure 1B). These patients were not allowed to undergo an oral feeding trial with an NGT inserted.

### 2.3. Interventions (Oral Diet Training with an NGT Inserted)

When the patient could safely swallow therapeutic food containing a certain viscosity without aspiration, the NGT was removed, and oral feeding was recommended. However, if the total amount of oral intake was insufficient, oral attempts combined with NGT feeding were recommended based on the results of the VFSS. Those patients were assigned to Group 2. In the beginning, an oral feeding trial was performed with an NGT inserted during swallowing therapy by an occupational therapist. After confirming the safety of oral feeding with an NGT inserted during swallowing therapy, the patients in Group 2 attempted an oral feeding trial in the ward at every mealtime before feeding through an NGT. The physicians supervised for the first few times. In these cases, oral feeding trials were permitted with a minimum dose of 100 mL at a time and a total amount of less than 500 mL per day for the first few days until the patient could safely swallow orally without complications. Only soft and thick therapeutic foods with a viscosity similar to that of yogurt, not liquids or thin fluids, were allowed during oral trials. The oral feeding trial volume gradually increased according to the patient’s condition. In patients with severe dysphagia who were unable to safely receive oral feeding, NGT reinsertion was recommended, and oral trials were prohibited (Group 3). All patients received a personalized dietary prescription according to the results of VFSS and received conventional dysphagia therapy. Follow-up VFSS was performed every 2 to 4 weeks, depending on the patient’s condition. Although dysphagia is a lifespan problem [19], our study focused on an adult population.

All patients who enrolled in this study during their hospitalization were admitted in the Department of Physical Medicine and Rehabilitation, University Hospital. Thus, each patient received comprehensive and multiple forms of rehabilitation therapy, including swallowing and speech therapy, as well as physical therapy, occupational therapy, and cognitive therapy, according to their condition.

## 3. Results

As shown in the flowchart (Figure 2), we reviewed the medical chart of 175 patients with prolonged dysphagia who were fed via an NGT for more than 4 weeks. A total of 129 patients were allowed to initiate an oral diet using therapeutic food after the VFSS. Among the patients, 37 (28.7%) required NGT reinsertion due to failure to achieve a sufficient oral feeding amount after NGT removal. These patients were assigned to Group 2, where thick foods were tried orally while the NGT was inserted. Of the 46 patients who required NGT reinsertion after the VFSS, 12 (26.1%) had no aspiration of thick or thin fluid during the VFSS with an NGT inserted. Therefore, these patients were also assigned to Group 2 and orally administered thick foods with an NGT inserted. Of the 49 patients in Group 2 who underwent oral trials with an indwelling NGT, 39 (79.6%) were transitioned to exclusive oral feeding. A transition period of 3–8 weeks was required for these patients to achieve full oral feeding and removal of the NGT. Among the remaining 10 patients, aspiration pneumonia occurred in 4 patients; otherwise, no further attempts were made because the amount of oral intake was too low.

The demographic characteristics of the patients in each group are shown in Table 1. The patients in Group 3 who were not allowed to attempt oral feeding trials with an NGT inserted were significantly older, had a longer duration of NGT feeding before VFSS evaluation, had a greater rate of tracheostomy, and had lower coughing ability than the patients in Groups 1 and 2.

Table 2 shows the changes in swallowing function in each group with and without an NGT inserted during the VFSS. For the patients in Group 1 who could receive oral feeding only, the FDS score was not different between the VFSS with or without an NGT inserted. The patients in Group 2 were eligible to undergo oral trials with an NGT inserted, and no significant differences were found in the FDS scores between the VFSS with or without an NGT inserted during the VFSS except for the oral phase of the FDS. There was no significant aspiration during the VFSS when an NGT was inserted. However, the patients in Group 3 who required prolonged NGT feeding without oral trials showed significant aspiration when they underwent VFSS with an NGT inserted. The amount of residue in the valleculae pouch and pyriformis sinuses was greater with an NGT inserted than without an NGT inserted. When comparing conditions with and without an NGT inserted, the FDS and PAS scores of patients without an NGT inserted were slightly and significantly improved compared to those with an NGT inserted.

As mentioned in Figure 1, the patients assigned to Group 2 did not undergo aspiration when performing VFSS with an NGT inserted, while the patients assigned to Group 3 underwent aspiration when performing VFSS with an NGT inserted. Comparing the results of the three groups without an NGT-inserted condition, patients in Group 3 had significantly higher FDS scores in both the oral and pharyngeal phases and higher aspiration scores in the PAS than those in the other groups (Table 2). The patients in Group 3 also had significantly lower coughing ability than patients who could undergo an oral trial with an indwelling NGT (Table 1).

## 4. Discussion

In the present study, we aimed to investigate the therapeutic effect of oral diet training in long-term indwelling NGT patients with chronic dysphagia. Our results demonstrated that the effect of long-term NGT insertion on swallowing differed according to various factors, and our study also quantitatively revealed the effect of successive oral feeding training with an NGT inserted in patients with long-term indwelling NGT. Most patients (79.6%) were able to obtain an adequate amount of nutrition only orally, and the NGT could be removed after oral diet training with an NGT inserted.

As mentioned previously, passing a swallowing test does not guarantee the end of NGT feeding. Our data showed that among the 129 patients who underwent NGT removal and who started an oral diet after the VFSS, 37 (28.7%) returned to NGT feeding again because they did not achieve enough nutrients through oral feeding after NGT removal. In most cases, the patients were able to obtain an adequate amount of nutrition orally after oral diet training with an NGT insertion state. And then, they finally could undergo NGT removal.

Furthermore, of the 46 patients who required NGT feeding after the VFSS, 12 (26.1%) were able to attempt oral diet training with an NGT inserted, and some of these patients achieved oral feeding exclusively after oral diet training. Patients with prolonged NGT insertion may lose oropharyngeal muscle swallowing function. For these patients, a transitional period involving training is required during the transition from NGT feeding to full oral feeding.

### 4.1. Factors Associated with Successive Oral Feeding

In this study, we attempted to identify factors that may be related to safe oral feeding with an indwelling NGT. Our results showed that in patients with sufficient coughing function and no aspiration with an NGT inserted during the VFSS, oral diet training would be a useful and safe procedure for swallowing treatment.

It is noteworthy that the patients in Group 3 who were unable to attempt oral feeding due to significant aspiration during the VFSS with an NGT inserted were significantly older and had a higher rate of tracheostomy than the patients in Groups 1 and 2 (Table 1).

Age could also be a significant factor in determining whether a patient can receive oral feeding. Previous studies revealed that an indwelling NGT had little effect on swallowing ability in young and healthy adults [20], whereas the opposite result was demonstrated in healthy older adults [21]. These studies showed an age-related effect on swallowing function [4,22]. Aging is directly related to the risk of multiple degenerative diseases and sarcopenic dysphagia [23].

The presence of a tracheostomy tube also negatively affects swallowing function. Several mechanisms have been proposed for swallowing dysfunction after a tracheostomy. The tethering of the larynx by the tracheostomy tube may result in decreased laryngeal elevation [24]. The pharyngeal pathway could be directly obstructed by the tube cuff [25]. Prolonged air diversion also causes desensitization of the larynx [26]. Furthermore, tracheostomy also negatively affects coughing ability [18].

According to our results and previous findings, important factors prior to attempting oral feeding with an NGT inserted are sufficient cough function, no aspiration during a VFSS with an NGT inserted, younger age, and no tracheostomy. These factors are associated with the successive achievement of oral feeding even when patients have a long-term indwelling NGT. Because prolonged NGT insertion is an independent and negatively influencing factor in stroke recovery [6], the NGT must be removed as soon as possible when a patient can safely consume an oral diet.

### 4.2. Effect of NGT on Swallowing Function

Previous studies have reported various effects of NGT on swallowing, but the conclusions remain controversial. Anatomically, the NGT occupies the space of the nasopharynx, oropharynx, and hypopharynx and can interfere with pharyngeal swallowing in patients with dysphagia [3]. NGT feeding may interfere with the movement of the hyoid bone during swallowing [4,27]. In healthy adults, the presence of an NGT is associated with the delayed initiation of maximal pharyngeal excursion. It also prolongs the opening time of the upper esophageal sphincter and total swallowing time [20]. A recent study demonstrated that the presence of an NGT negatively impacted swallowing function in elderly stroke patients. Swallowing evaluations were significantly different before and after NGT removal [4,28].

The opposite results were shown in another study in which no significant differences were found in swallowing function before and after NGT removal [29]. A review article concluded that the risk of aspiration from small amounts of liquid did not differ significantly before and after NGT removal, regardless of dysphagia or general functional level [27].

Why are NGT effects different according to these studies? Because dysphagia patients have various etiologies and exhibit multiple, diverse clinical manifestations and recovery processes. Our results also revealed that the effect of long-term NGT insertion on swallowing differs according to various factors. Thus, we triaged the patients and managed them differently according to their swallowing ability.

As previously described, multiplex factors appear to be associated with swallowing function and the ability to achieve full oral feeding after long-term NGT insertion, and it seems difficult to generalize and draw conclusions about the impact of long-term NGT insertion on swallowing. Individual evaluation and multidisciplinary approaches are needed for chronic dysphagia patients to achieve successful oral feeding.

### 4.3. Therapeutic Effect of Oral Diet Training with an Indwelling NGT

Generally, the goals of dysphagia treatment focus on safe oral diets and adequate nutritional intake, which include oral motor facilitation, electrical stimulation, and muscle strength training related to swallowing and breathing [30,31].

Oral motor therapy is a traditional effective dysphagia therapy [30,32] that consists of direct manual stroking, active oral motor exercises, and passive sensory stimulation. Direct oral diet training can certainly achieve these oral motor facilitation effects, and furthermore, it can induce the coordination of swallowing muscles.

Direct oral training may also have the effect of strengthening the pharyngeal muscles by stimulating the pharyngeal walls when swallowing food. It affects pharyngeal efferent nerves and triggers swallowing reflexes and pharyngeal wall movement. Long-term NGT indwelling may cause patients to gradually lose the ability to chew and swallow. And in patients with pharyngeal weakness due to neurogenic or degenerative dysphagia, the mechanical effect of an NGT may negatively affect swallowing [4,18]. However, after oral diet training, patients who had improved pharyngeal weakness or recovered dysphagia were unaffected by the mechanical effect of NGT on swallowing.

A previous study revealed that one of the factors associated with the achievement of oral intake ability in stroke patients was starting oral intake early [33]. Our protocol could help patients start oral intake combined with an indwelling NGT early, resulting in better functional outcomes. Our oral diet training also helps patients gain confidence that they can eat orally by repeatedly practicing food-swallowing not only in the treatment room but also in the ward.

Based on our results, we propose that if the patient is a proper, potential candidate for NGT removal, our direct oral feeding training with an NGT inserted can accelerate NGT removal and help patients achieve full oral feeding. To date, direct oral feeding training attempts with an NGT inserted have not been evaluated, and our trial is the first preliminary study.

### 4.4. Percutaneous Endoscopic Gastrostomy Tube Insertion

In terms of PEG tube insertion, there are no strict guidelines on who is the best candidate or when is the best time. Current guidelines recommend PEG tube insertion in cases of prolonged dysphagia of more than 4–6 weeks [8,9,10], and many patients with prolonged NGT insertion do not want to undergo PEG tube insertion for various reasons [11]. Some of the included patients had an indwelling NGT for a period exceeding 36 months [11].

In our study, among the 175 patients who had a prolonged indwelling NGT, 44 (25.1%) could not undergo NGT removal during the study period. We suggest that if a patient is older and has no expectation of improvement in oral feeding due to brain lesions of the central pattern generator involved in the swallowing function or advanced stages of dementia or Parkinsons disease, it may be better to insert a PEG tube early to support adequate nutrition, improve the patient’s quality of life, and prevent further complications. A recent review article recommended PEG tube insertion before weight loss during disease to provide adequate nutrition [9]. Even with PEG tube insertion, our oral diet training protocol can be applied if a patient’s condition improves and there are appropriate indications for our protocol.

### 4.5. Limitations

Our study was designed as a retrospective medical chart review, so we were unable to conduct a randomized controlled trial of oral diet training. In addition, this pilot study included a small number of patients, and we could not divide patients according to brain lesion or disease characteristics, such as acquired, degenerative, or progressive disease. And because this study was a retrospective study, the patient’s cognitive function, which may affect swallowing function, could not be evaluated. Patients had different indwelling NGT periods, and long-term follow-up VFSS could not be performed for all patients, which may have affected the results. Therefore, further studies involving more participants and well-designed randomized controlled trials with longer follow-up periods are needed.

## 5. Conclusions

This study investigated the therapeutic effect of oral diet training with an NGT inserted in patients with chronic dysphagia. Our results quantitatively revealed the successive effects of oral feeding training with an NGT inserted in patients with a long-term indwelling NGT. We also provided the appropriate indications for direct oral training. Direct oral training with an NGT inserted has therapeutic effects on oral motor facilitation, swallowing muscle coordination, and pharyngeal wall stimulation.

As chronic dysphagia patients exhibit different recovery processes associated with multiple factors, frequent evaluation and appropriate personalized management of dysphagia symptoms are very important strategies. We suggest that if a patient is a proper candidate for NGT removal, our method of direct oral feeding training with an NGT inserted could be a useful therapeutic strategy during the transitional period from long-term NGT feeding to successful oral feeding.

## Figures and Tables

**Figure 1 nutrients-16-02424-f001:**
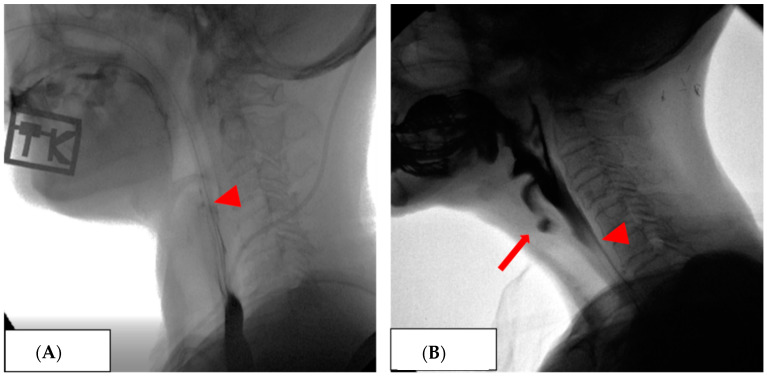
The findings of the VFSS with an NGT inserted. (**A**): Patients assigned to Group 2 had no aspiration when oral swallowing was attempted with an NGT inserted (arrowhead) during the VFSS (Appendix A). (**B**): Patients assigned to Group 3 showed aspiration (arrow) when they tried to swallow orally with an NGT inserted (arrowhead) (Appendix A).

**Figure 2 nutrients-16-02424-f002:**
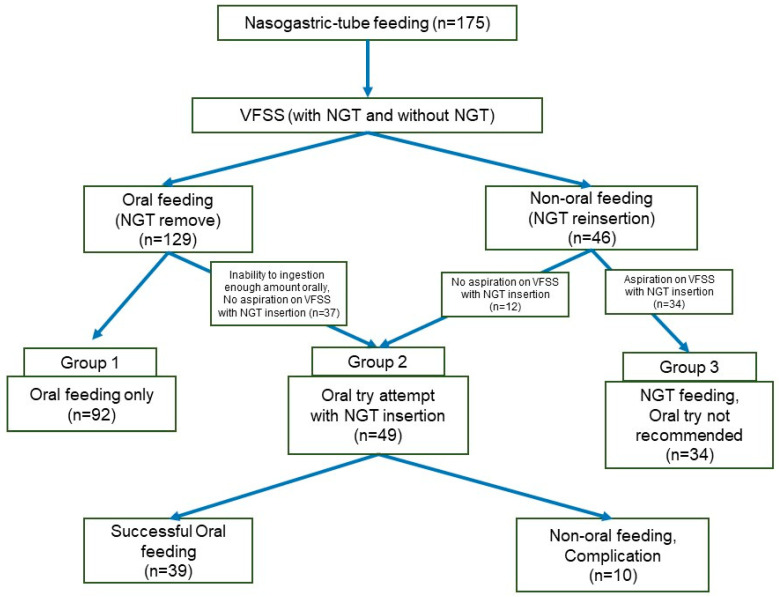
Flowchart of the study.

**Table 1 nutrients-16-02424-t001:** Demographic characteristics of patients.

Parameters	Group 1 (*n* = 92)	Group 2 (*n* = 49)	Group 3 (*n* = 34)	*p*-Value
Age	74.7 ± 11.2	68.2 ± 24.5	82.4 ± 10.9	0.037
Gender (M/F)	58/34	22/27	18/16	
Cause of dysphagia				
Stroke	66 (71.7%)	34 (69.4%)	20 (58.8%)	
Hemorrhagic	32 (34.8%)	8 (16.3%)	14 (41.2%)	
Ischemic	34 (37.0%)	26 (53.1%)	6 (17.6%)	
Traumatic brain injury	12 (13.0%)	8 (16.3%)	0	
Parkinsons	0	2 (4.1%)	5 (14.7%)	
ALS	0	1 (2.0%)	1 (3.0%)	
Myopathy	3 (3.3%)	2 (4.1%)	0	
Dementia	8 (8.7%)	0	6 (17.6%)	
Others (medical condition)	3 (3.3%)	2 (4.1%)	2 (5.9%)	
L-tube feeding duration (days)	51.6 ± 35.4	66.1 ± 40.6	79.1 ± 14.8	0.01
Tracheostomy	2 (2.2%)	13 (26.5%)	19 (55.9%)	0.001
PCF	168.0 ± 78.5	138.7 ± 28.5	84.4 ± 47.2	0.001
MBI	38.3 ± 12.6	21.4 ± 9.0	19.6 ± 14.5	0.951

Values are number (%). ALS: amyotrophic lateral sclerosis, PCF: peak cough flow, and MBI: modified Barthel index.

**Table 2 nutrients-16-02424-t002:** Changes in swallowing and coughing functions in each group with and without the NGT.

	Group 1 (*n* = 92)			Group 2 (*n* = 49)			Group 3 (*n* = 34)			
	L-tube (+)	L-tube (−)	*p*-Value	L-tube (+)	L-tube (−)	*p*-Value	L-tube (+)	L-tube (−)	*p*-Value	*p*-Value (Compare Three Group without NGT)
FDS (total, 0–100)	41.5 ± 13.1	44.7 ± 13.8	0.974	50.2 ± 11.8	48.2 ± 12.7	0.772	63.1 ± 14.8	52.3 ± 10.9 *†	0.03	0.012
FDS (oral, 0-)	7.2 ± 2.2	3.9 ± 3.2	0.515	10.1 ± 5.4	8.7 ± 5.5 *	0.032	11.1 ± 5.2	10.7 ± 5.0	0.572	0.001
FDS (pharyngeal, 0-)	37.2 ± 10.2	37.1 ± 11.2	0.968	40.2 ± 9.7	39.1 ± 10.6	0.951	49.1 ± 11.2	42.5 ± 8.8 *	0.002	0.025
FDS (vallecular residue, 0–12)	5.3 ± 3.8	5.5 ± 4.2	0.829	6.2 ± 4.1	5.8 ± 3.9	0.671	9.5 ± 3.2	6.5 ± 2.0 *†	0.001	0.037
FDS (piriformis sinus residue (0–12)	3.2 ± 3.3	3.0 ± 3.3	0.695	4.4 ± 2.6	4.1 ± 4.5	0.903	7.3 ± 4.3	7.0 ± 3.6 †	0.725	0.018
PAS (0–8)	2.1 ± 0.6	1.2 ± 0.5	0.711	3.7 ± 2.8	1.6 ± 1.2	0.513	6.4 ± 2.5	5.8 ± 3.7 †	0.397	0.001

Values are scores. FDS: functional dysphagia scale, PAS: penetration aspiration scale, *: *p* < 0.05 according to paired *t*-test before and after NGT removal within groups, †: *p* < 0.05 according to one-way analysis of variance (ANOVA), and comparisons of three group without NGT.

## Data Availability

The data of this study are available from the corresponding author on reasonable request.

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
