# Peer review of "The Effect of Oral Diet Training in Indwelling Nasogastric Tube Patients with Prolonged Dysphagia"

_nutrients, 2024, doi:10.3390/nu16152424_

Round 1

Reviewer 1 Report

Comments and Suggestions for Authors

Please consider the following changes to your manuscript:

- You need to mention that dysphagia is a lifespan problem, and that it can also impact on children, with the need for non-oral feeding methods on occasion. State that although you recognise this, your paper specifically focuses on an adult population. Please mention this and cite: van den Engel-Hoek, L., et al. (2017). Pediatric feeding and swallowing rehabilitation: An overview. Journal of pediatric rehabilitation medicine10(2), 95-105.

-You have mentioned the different types of aetiologies in Table 1, but I think you need to expand in your Introduction discussion about onset of dysphagia in your population, e.g. acquired versus degenerative problems. This is important as different types of problems respond to interventions to remediate dysphagia differently according to whether the conditions are acquired or progressive, and if cognition is impacted or not. 

- I think you need to mention that you used the PAS in the 2.1 section when you actually mention that you undertook VFSS. 

- Did you use speech and language therapists at all? If so, when. If you didn't, why not?

- I think for the purposes of your readership , I would change to term "physiatrist" and use "physician" instead.

-What was the actual rationale for the oral trials, and what was "oral diet training"? Oral motor dysphagia work is wide ranging and in some cases controversial. You should briefly discuss the challenges of oral motor interventions, and in the Discussion, recognise that this is an ongoing area of medicine in which further studies are urgently needed. 

- Did participants keep eating orally on discharge home? What is your follow-up plan?

- Did you have any infection issues with your NGTs? It can be a significant problem for some institutions, and it needs to be mentioned. 

Author Response

  1. You need to mention that dysphagia is a lifespan problem, and that it can also impact on children, with the need for non-oral feeding methods on occasion. State that although you recognize this, your paper specifically focuses on an adult population. Please mention this and cite: van den Engel-Hoek, L., et al. (2017). Pediatric feeding and swallowing rehabilitation: An overview. Journal of pediatric rehabilitation medicine10(2), 95-105.

Reply: Thank you for your valuable advice.

As you mentioned, we completely agree that dysphagia is a lifespan problem, from infant to the geriatrics. We cited this reference in the intervention section (2-1. participant). We appreciate your kindness.

  1. You have mentioned the different types of etiologies in Table 1, but I think you need to expand in your Introduction discussion about onset of dysphagia in your population, e.g. acquired versus degenerative problems. This is important as different types of problems respond to interventions to remediate dysphagia differently according to whether the conditions are acquired or progressive, and if cognition is impacted or not. 

Reply: Thank you for your very thorough comment.

I absolutely agree with your valuable opinion that dysphagia has various etiologies, undergo different recovery processes, and is also influenced by cognitive function and disease type. (acquired, degenerative or progressive). All enrolled patients were elderly (64 – 96 years old), and therefore most of them had acquired (Stroke, TBI et al), and degenerative or progressive (Dementia, Parkinsonism, ALS et al) diseases. And NGT feeding duration marks the onset of dysphagia (or dysphagia duration) in the Table 1.

We noted that “As patients with dysphagia have various etiologies and undergo different recovery processes, multiple and complex factors are associated with successful exclusive oral feeding. Frequent evaluation and appropriate management of dysphagia symptoms are very important strategies” in the introduction section.

In facts, because this study was a retrospective study, the patients’ cognitive function, which may affect swallowing function, was not evaluated. And our pilot study included a small number of patients, we could not divide patients according to brain lesion or disease characteristics such as acquired, degenerative or progressive. I’m so sorry. That’s our limitation. Thus, we explained it in the discussion limitation section 4-5.

“This pilot study included a small number of patients, and we could not divide patients according to brain lesion or disease characteristics such as acquired, degenerative or progressive. And because this study was a retrospective study, the patient’s cognitive function, which may affect swallowing function, cannot be evaluated.”

Our results also revealed that the effect of long-term NGT insertion on swallowing definitely differs according to various factors. Thus, we triaged the patients and managed them differently according to their swallowing ability.

We mentioned these contents in the introduction, results and discussion section.

“As patients with dysphagia have various etiologies and undergo different recovery processes, multiple and complex factors are associated with successful exclusive oral feeding. Frequent evaluation and appropriate management of dysphagia symptoms are very important strategies”

“As previously described, multiplex factors appear to be associated with swallowing function and the ability to achieve full oral feeding after long-term NGT insertion, and it seems difficult to generalize and draw conclusions about the impact of long-term NGT insertion on swallowing. Individual evaluation and multidisciplinary approaches are needed for chronic dysphagia patients to achieve successful oral feeding.”

In a later discussion section, 4-4, we noted that patients who are very elderly, and have severe dementia or have parkinsonism such as degenerative or progressive dysphagia may benefit from performing PEG instead of oral feeding.

“We suggest that if a patient is older and has no expectation of improvement in oral feeding due to brain lesions of the central pattern generator involved in swallowing function or advanced stages of dementia, it may be better to insert a PEG tube early to support adequate nutrition, improve the patient’s quality of life and prevent further complications”

Once again, we appreciate your valuable comments.

  1. I think you need to mention that you used the PAS in the 2.1 section when you actually mention that you undertook VFSS. 

Reply: Thank you for your helpful comment. We mentioned PAS in the 2.1 section. And PAS was explained in detail in this section.

There are many tools available to describe swallowing function based on the results of VFSS (PAS, DOSS, FDS, VDS et al.) Among them, the reason for using PAS is that it is simple and clearly shows the presence of penetration or aspiration in numbers.

We have edited the Methods section based on your helpful comment. Thank you again.

2.1 Evaluations (VFSS) FDS, PAS

2.2 Patient grouping   

2.3 Intervention (Oral diet training with an NGT inserted)

  1. Did you use speech and language therapists at all? If so, when. If you didn't, why not?

Reply: Thank you for your very important question.

All patients who enrolled in this study received swallowing and speech therapy according to their condition at our university hospital.   

We added these to the method section

“All patients who enrolled in this study were conducted during their hospitalization in the department of Physical Medicine and Rehabilitation, university hospital. Thus, each patient received comprehensive and multiple rehabilitation therapy including swallowing and speech therapy, as well as physical therapy, occupational therapy and cognitive therapy.”

In addition, “In the beginning, an oral feeding trial was performed with an NGT inserted state during swallowing therapy by an occupational therapist. After confirming safety oral feeding with an NGT inserted state during the swallowing therapy, the patients in Group 2 attempted an oral feeding trial in the ward at every meal time before feeding through NGT. The physicians initially supervised first few times.” We also added it in the methods section 2-3. 

We sincerely appreciate your very, very important comments.

  1. I think for the purposes of your readership, I would change to term "physiatrist" and use "physician" instead.

Reply: Thank you for your thorough advise. We changed “physiatrist” to “physicians”.

“Physiatrist” means a physician who specializes in physical medicine and rehabilitation. However, I think this term might be unfamiliar to reader. Thank you again for your advice.

  1. What was the actual rationale for the oral trials, and what was "oral diet training"? Oral motor dysphagia work is wide ranging and in some cases controversial. You should briefly discuss the challenges of oral motor interventions, and in the Discussion, recognize that this is an ongoing area of medicine in which further studies are urgently needed. 

Reply: Thank you for your very important question. “Direct oral diet (feeding) training” is a very important conventional training in dysphagia therapy. However, conventional direct oral diet training was performed using small amounts of yogurt or therapeutic food only in the rehabilitation treatment room without NGT inserted state.

Conventional oral training

This study’s protocol

Oral diet (feeding) training

NGT insertion

Without an NGT inserted state

With an NGT inserted state

Oral trial place

Only in the swallowing therapy room

Not in the ward.

At first, in the swallowing therapy room.

Sequentially, at each meal time in the ward.

Oral trial amount

Small amount of therapeutic food

Initially, a minimum dose of 100 ml at a time and a total amount less than 500 ml per day is allowed for the first few days until the patient could safely swallow orally without complications.

Then, oral feeding amount was gradually increased according to the patient’s condition.

The most important thing that distinguishes our study from other study is “direct oral diet (feeding) trial with an NGT inserted state.” You can check it with Fig.1 and appendix video files.

Our results revealed that the effect of long-term NGT insertion on swallowing definitely differs according to various factors. Thus, we triaged the patients and managed them differently according to their swallowing ability based on the results of VFSS.

Patients assigned to Group 2 had no aspiration when oral swallowing was attempted with an NGT inserted (arrow head) during the VFSS (Fig. 1-A). Patients assigned to Group 3 showed aspiration (arrow) when they tried to swallow orally with an NGT inserted (arrow head) in the Fig 1-B.

 Patients assigned to the Group 2 attempted a “direct oral feeding trial with an NGT inserted state” in the swallowing therapy room and subsequently in the ward every meal time. They gradually increase the oral feeding amount.  

Actual rationale: We summarized the effect of oral diet training with an indwelling NGT in the discussion section 4-3.

Oral diet (feeding) training with NGT inserted state may have these effects;

#1. OMF: Direct oral diet training with NGT inserted can achieve oral motor facilitation effects (active oral motor exercises, passive sensory stimulation),

#2. It can induce the coordination of swallowing muscles. 

#3. It also has the effect of strengthening the pharyngeal muscles by stimulating the pharyngeal wall when swallowing of food. 

#4. Direct oral diet trial can start oral intake early. Long-term NGT indwelling may cause patients to gradually lose the ability to chew and swallow. Our protocol could help patients start oral intake combined with an indwelling NGT early.

#5. Our protocol (Direct oral diet training) helps patients gain confidence that they can eat by orally by repeatedly practicing food swallowing not only in the treatment room, but also in the ward.

  1. Did participants keep eating orally on discharge home? What is your follow-up plan?

Reply: Thank you for your very important questions. As we mentioned in Fig 2. (Flowchart) 39 patients in the Group 2 (chronic dysphagia patients who had NGT feeding for more than 4 weeks) could remove NGT and exclusively feeding orally.

Of the 49 patients in Group 2 who underwent oral trials with an indwelling NGT, 39 (79.6%) were transitioned to exclusive oral feeding. A transition period of 3-8 weeks was required for these patients to achieve full oral feeding and removal of the NGT. They keep eating orally on discharge.

All included patients were admitted in our rehabilitation department. And after treatment, they were discharged to home, a nursing center, or other rehabilitation hospitals. Patients were followed up at OPD regularly as scheduled. In addition, VFSS was reassessed if patients had aspiration symptoms or other newly onset dysphagia symptoms.

  1. Did you have any infection issues with your NGTs? It can be a significant problem for some institutions, and it needs to be mentioned. 

Reply: Thank you for your thorough comment. As you mentioned, infection related to NGT would be a significant problem. We totally agree with your opinion.

We wrote introduction the complication of prolonged NGT insertion.

“Long-term NGT placement can also cause complications such as gastroesophageal reflux or aspiration pneumonia, nasal wing lesions, and chronic sinusitis”

And, in the result section, 10 patients who attempt oral trials with NGT insertion state could not achieve exclusive oral feeding. They were kept NGT feeding without oral trial or PEG insertion. Among these 10 patients, 4 patients had suffered from aspiration pneumonia relating oral diet feeding or other causes. We mentioned as you recommended.

Once again, we completely agree that prolonged NGT causes various infectious problems.

Please receive our revised manuscript for consideration for publication in Nutrients.

Thank you for giving us the opportunity to submit our revised manuscript.

Reviewer 2 Report

Comments and Suggestions for Authors

Therapeutic effect of oral diet training….

Retrospective observational study on the results of a program devoted to re-educate oral feeding skills in patients with dysphagia and an indwelling NG tube.

Comments to the authors

1.      I’d suggest to change the title as it is not a therapeutic effect but Effect or Results.

2.      Introduction

a.      Line 40. Dysphagia is not an indication for PN.

b.      NGT is in fact enteral nutrition (line 41). Please modify the sentence.

c.       The sense of the last sentence in this paragraph is not clear. The worse clinical outcome in patients with prolonged dysphagia is not because of carrying an NG tube.

3.      Material and methods

a.      VFSS appears first time in line 74, please write in the complete form

b.      What’s the meaning “while the NGT was insulted”? (line 84)

c.       Physiatrists = physical therapists? (line 86)

d.      The meaning “therapeutic food” is quite different that the one use by the authors.

Comments on the Quality of English Language

--

Author Response

Comments to the authors

  1. I’d suggest to change the title as it is not a therapeutic effect but Effect or Results.

Reply: Thank you for your valuable recommend. We changed the title from “Therapeutic effect” to “Effect” based on your recommendation. Thanks again.

  1. Introduction
  2. Line 40. Dysphagia is not an indication for PN.
  3. NGT is in fact enteral nutrition (line 41). Please modify the sentence.
  4. The sense of the last sentence in this paragraph is not clear. The worse clinical outcome in patients with prolonged dysphagia is not because of carrying an NG tube.

Reply: We sincerely appreciate your thoughtful comments. I admit that my expressions were vague and inappropriate. I’m so sorry about that.

We have revised this introduction paragraph according to your valuable recommendations.

Patients with severe swallowing difficulties usually use compensatory feeding method such as nasogastric tube (NGT) feeding or percutaneous endoscopic gastrectomy (PEG). NGT feeding may be an appropriate alternative type of enteral feeding method for patients who are unable to obtain adequate nutrition via oral route. However, some patients are fed using an NGT for relatively long periods, which may have negative impact on their swallowing function. Long-term NGT indwelling may cause patients to gradually lose the ability to chew and swallow. As a result, the stimulation the brain receives decreases and in the incidence of cognitive impairment increases. Furthermore, prolonged dysphagia is strongly associated with poor functional outcomes after brain injury.”

If there are any inappropriate expressions, please don’t hesitate to let us know.

  1. Material and methods
  2. VFSS appears first time in line 74, please write in the complete form
  3. What’s the meaning “while the NGT was insulted”? (line 84)
  4. Physiatrists = physical therapists? (line 86)
  5. The meaning “therapeutic food” is quite different that the one use by the authors.

Reply: Thank you for your valuable recommend. We edited these contents (a, b, c, d) according to your recommendations.

  1. We wrote full-term of VFSS in the manuscript. Thank you so much for checking my shortcomings.
  2. “While the NGT was insulted” It means that VFSS was conducted with NGT was insulted. Usually, swallowing study such as VFSS was done without NGT, but we conducted VFSS with NGT and without NGT.

You can see the Fig. 1 that patients were evaluated VFSS with NGT insertion (“arrow head” indicate NGT).

  1. “Physiatrist” means a “Physician who specializes in Physical Medicine and Rehabilitation”,

We changed physiatrists to the “physicians who specialize in physical medicine and rehabilitation” and “physicians”. All authors are rehabilitation medicine doctors. However, I think this term (Physiatrist) might be unfamiliar to reader. Thank you for your helpful advice.

  1. ”Therapeutic food” in our manuscript means soft and thick food with a viscosity similar to that of yogurt, not liquids or thin fluids. It’s a “thick food” for dysphagic patients.

The most important thing about therapeutic food is the thickness or viscosity of the food. At our hospital, the ‘hospital nutrition team’ prepares “therapeutic thick food” for dysphagic patients and supplies it to patients. We called this “Therapeutic food”.  

We have changed the “therapeutic food” to the “thick food”

Thank you again for your important advice.  

Please receive our revised manuscript for consideration for publication in Nutrients.

Once again, thank you for giving us the opportunity to submit our revised manuscript.

Round 2

Reviewer 1 Report

Comments and Suggestions for Authors

Thank-you for your thorough work to address review comments. The manuscript now reads well and I have no further comments.